



# Using nutation-frequency-selective pulses to reduce radio-frequency field inhomogeneity in solid-state NMR

Kathrin Aebischer[1], Nino Wili[1], Zdeněk Tošner[2], Matthias Ernst[1]

[1]Physical Chemistry, ETH Zürich, Vladimir-Prelog-Weg 2, 8093 Zürich, Switzerland

[2]Department of Chemistry, Faculty of Science, Charles University, Hlavova 8, 12842 Prague 2, Czech Republic

*Correspondence to*: Matthias Ernst (maer@ethz.ch)

**Abstract.** Radio-frequency (rf) field inhomogeneity is a common problem in NMR which leads to non-ideal rotations of spins in parts of the sample. Often, a physical volume restriction of the sample is used to reduce the effects of rf-field inhomogeneity especially in solid-state NMR where spacers are inserted to reduce the sample volume to the centre of the coil. We show that

band-selective pulses in the spin-lock frame can be used to apply $B_1$-field selective inversions to spins that experience selected parts of the rf-field distribution. Any frequency band-selective pulse can be used for this purpose but we chose the family of I-BURP pulses (H. Geen, R. Freeman, Band-Selective Radiofrequency Pulses, J. Magn. Reson. 93 (1991) 93–141) for the measurements demonstrated here. As an example, we show that the implementation of such pulses improves homonuclear frequency-switched Lee-Goldburg decoupling in solid-state NMR.

## 1 Introduction

Radio-frequency (rf) inhomogeneity is one of the experimental imperfections in solid-state NMR experiments that is almost unavoidable and often leads to deterioration of the performance of pulse sequences. Especially in small magic-angle spinning (MAS) probes used in high-resolution solid-state NMR where the solenoid coil is close to the sample space, significant rf-field inhomogeneity is observed over the sample volume (Tosner et al., 2017). The magnitude of the rf-field inhomogeneity can be

characterized by a simple nutation experiment. Characterizing the full spatial distribution of the rf-field amplitude over a rotor requires single- or triple-axis gradients for imaging (Guenneugues et al., 1999). Alternatively, the rf-field distribution over the rotor can be measured by a ball-shift measurement (Paulson et al., 2004) or calculated using numerical simulations of the coil and rf circuit (Tosner et al., 2017; 2018). Simulations and measurements show that typical MAS solid-state NMR probes have large rf-field distributions along the rotor axis and in addition along the radial dimension. Such rf-field inhomogeneity often

manifests itself in the spectrum as reduced signal intensity in polarization-transfer experiments (Nishimura et al., 2001), in broadened lines in decoupling experiments (Vega, 2004) or in spatial selectivity in cross-polarization experiments (Gupta et al., 2015).

Experimentally reducing the rf-field inhomogeneity is often achieved through sample restriction by inserting spacers into the upper and lower part of the rotor and filling the sample in the central part of the rotor. However, since there is also significant



radial rf-field inhomogeneity (Tosner et al., 2017) in MAS NMR probes, even very thin slices of samples still show a significant distribution of rf-field amplitudes. Sometimes, even spherically restricted samples are used inside the cylindrical rotors especially in high-resolution MAS (Lindon et al., 2009). In principle, magnetic-field gradients also allow the restriction of the sample space (Charmont et al., 2000) along the axis of the rotor if a magic-angle gradient is used. As there are very few probes in solid-state NMR that have gradient capability, this approach is not suitable for widespread application. Alternatively, radio-

frequency field selective pulses can be used to achieve sample restriction. This was demonstrated some years ago (Charmont et al., 2002), however, the numerically optimized radio-frequency selective pulses showed many sidebands especially in the low rf-field region (Charmont et al., 2002).

In this publication, we propose a simpler approach to implement amplitude selective pulses in the spin-lock frame by using a modulation of the pulses that is resonant with the spin-lock field. In such an implementation, any frequency-selective pulses

(Emsley, 2007) that have been designed for chemical-shift selection can be used. In our work, we have chosen the I-BURP class of band-selective inversion pulses (Geen and Freeman, 1991) that show a very sharp inversion profile. Similar approaches using resonant rf irradiation in nested rotating frames have been reported before in liquid-state NMR (Grzesiek and Bax, 1995), in recoupling experiments in solid-state NMR (Khaneja and Nielsen, 2008; Straasø et al., 2009) and in EPR (Wili et al., 2020).

## 2 Theory

In NMR, radio-frequency pulses are implemented as resonant rf irradiation orthogonal to the static magnetic field that leads to a static magnetic field in the rotating frame (Ernst et al., 1990). The Hamiltonian in the laboratory frame is given by

$$\mathcal{H}(t) = \mathcal{H}_Z + \mathcal{H}_{rf}(t) = \omega_0 I_z + 2\omega_1(t)\cos(\omega_{rf}t + \varphi(t)) I_x \qquad (1)$$

the sum of the Zeeman interaction and the time-dependent radio-frequency term describing a linear-polarized resonant radio-frequency irradiation. After transformation to the usual rotating frame (with the modulation frequency of the radio frequency),

we obtain two terms. One has a zero frequency and is static in the rotating frame while the second one rotates at twice the frequency and is usually neglected. It can, however, give rise to Bloch-Siegert effects (Bloch and Siegert, 1940). Therefore, we obtain a first-order rotating-frame Hamiltonian of the form

$$\mathcal{H}'(t) = (\omega_0 - \omega_{rf})I_z + \omega_1(t)(\cos\varphi(t) I_x + \sin\varphi(t) I_y) \qquad (2)$$

Note, that the factor of two in front of $\omega_1(t)$ in Eq. (1) is just introduced for convenience to obtain an amplitude of $\omega_1(t)$ in

the rotating frame. Application of circular-polarized rf is also possible and would eliminate this factor of two but is rarely implemented in NMR. Using an appropriate phase or amplitude modulation will allow us to implement frequency band-selective pulses on the Larmor-frequency axis (Emsley, 2007). Assuming that $\omega_0 - \omega_{rf}$ is small (near resonance irradiation), we can now apply a strong spin-lock field along the x axis and use an orthogonal resonant field to apply pulses in the spin-lock frame. The Hamiltonian for such an irradiation in the usual rotating frame would be


$$\mathcal{H}'(t) = \omega_1 I_x + 2\omega_2(t)\cos(\omega_{mod}t) I_y . \qquad (3)$$



Note the similarity of Eq. (3) to Eq. (1). Transforming into an interaction frame with $\omega_{mod}I_x$ and neglecting again the counter-rotating part leads to a first-order interaction-frame Hamiltonian of the form

$$\mathcal{H}''(t) = (\omega_1 - \omega_{mod})I_x + \omega_2(t)I_y \tag{2}$$

where $\omega_1$ might be broadly distributed due to rf-field inhomogeneity. We can now choose $\omega_2(t)$ such that it corresponds to a frequency-band selective pulse and set $\omega_{mod} = \omega_1$ to obtain a radio-frequency field selective pulse, i.e. a pulse that is selective in $\omega_1$. Again, we can implement any arbitrary shaped pulse that was designed for band selection in the chemical-shift space (Emsley, 2007). Figure 1 illustrates how such a selective pulse in the spin-lock frame is implemented for the example of an I-BURP-2 pulse (Geen and Freeman, 1991) that has been used in our experiments.

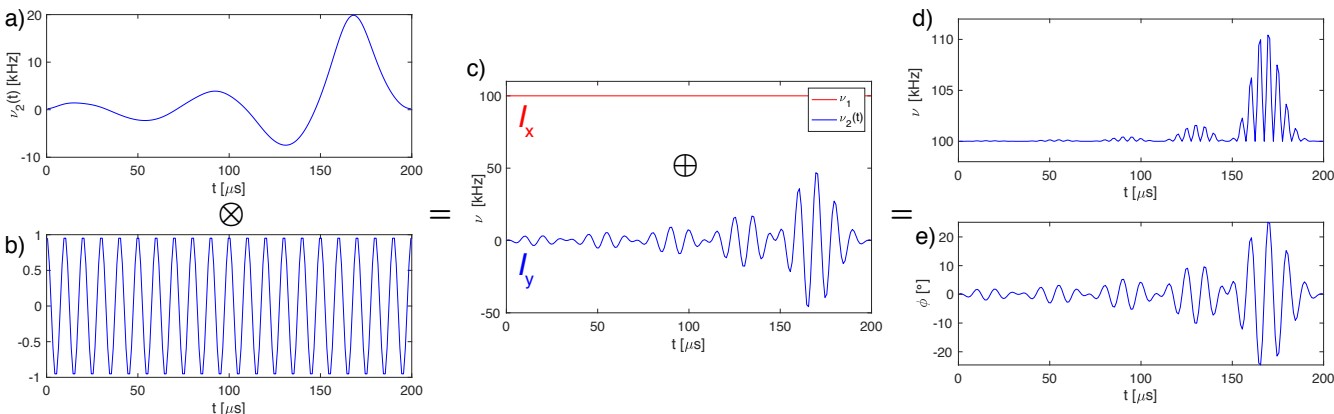

**Figure 1:** Generating the I-BURP-2 pulse in the spin-lock frame with 100 kHz spin-lock field, 200 μs pulse length, 100 kHz modulation frequency and a time resolution of 1 μs. a) Pulse shape of the I-BURP-2 pulse which is multiplied with the b) frequency modulation and leads to the c) y component (blue) of the rf field. The spin lock is the x component (red) of the rf field. The two components are combined and result in d) the amplitude and e) phase of the final pulse shape that can be used in the experiment. The time resolution for the shapes was usually set to 1 μs.

By changing the modulation frequency, we can select different parts of the rf-field distribution in the probe and by changing the pulse length we can adjust the width of the selected region. The bandwidth of the I-BURP-2 pulse is roughly $4/\tau_p$ where $\tau_p$ is the length of the pulse. The pulse shown in Fig. 1 with $\tau_p = 200$ μs has, therefore, an excitation bandwidth of approximately 20 kHz. If the amplitude of the pulse in the spin-lock frame is not much lower than the amplitude of the spin lock itself (very broad excitation band width), Bloch-Siegert type (Bloch and Siegert, 1940) phenomena might become visible. In this case, a pulse using circular polarized irradiation in the spin-lock frame can, in principle, be used to avoid these problems.

## 3 Numerical Simulations

Numerical simulations of the performance of the I-BURP-2 pulses in the spin-lock frame were carried out using the GAMMA spin-simulation environment (Smith et al., 1994). Sweeping the spin-lock field from 10 to 150 kHz, using a modulation frequency of 100 kHz, and a pulse length of 1 ms (Fig. 2a) and 200 μs (Fig. 2b), respectively, we have simulated the inversion efficiency of the spin-locked magnetization. Figure 2 shows that the bandwidth is indeed roughly 4 kHz (Fig. 2a) and 20 kHz



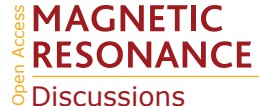

(Fig. 2b) and outside this band there are only very small artefacts visible. The line in black shows the profile assuming an ideal rf-field amplitude for the I-BURP-2 pulse ($\nu_2 = 1/\tau_p$) corresponding to a perfect inversion pulse independent of the value of $\nu_1$ while for the blue line the rf-field amplitude was scaled ($\nu_2 = (1/\tau_p)(\nu_1/\nu_{mod})$) leading to flip-angle deviation in the inversion pulse if the spin lock amplitude ($\nu_1$) is different from the modulation frequency ($\nu_{mod}$). There are only small

90 differences between the two simulations, indicating a good compensation of rf-field amplitude errors in the I-BURP-2 pulse (Geen and Freeman, 1991). This clearly shows the excellent radio-frequency amplitude selectivity of the I-BURP-2 pulse in the spin-lock frame. Changing the pulse length will allow us to adjust the bandwidth of the selected area. In principle, any selective pulse used in NMR can be implemented in this scheme.

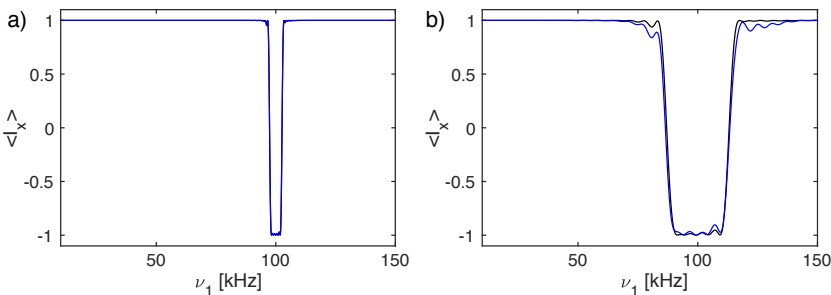

95 **Figure 2:** Plot of the expectation value $\langle I_x \rangle$ as a function of the rf-field amplitude $\nu_1$ in the range of 10 to 150 kHz under an I-BURP-2 pulse in the spin-lock frame using ideal rf-field amplitudes (black) and scaled rf-field amplitudes (blue). The modulation frequency was set to $\nu_{mod} = 100$ kHz and the pulse had a length of a) 1 ms and b) 200 μs corresponding to an inversion range of about 4 and 20 kHz, respectively. One can clearly see the inversion band around the 100 kHz modulation frequency with virtually no artefacts outside the inversion band. The black line corresponds to an ideal rf-field amplitude of the I-BURP-2 pulse of $\nu_2 = 1/\tau_p$ while the blue line uses $\nu_2 = (1/\tau_p)(\nu_1/\nu_{mod})$.

## 100  4 Experimental Results and Discussion

The experimental implementation of such pulses was tested by combining an I-BURP-2 inversion pulse in the spin-lock frame with a 2D nutation experiment. A schematic representation of the pulse sequence is shown in Fig. 3. The I-BURP-2 pulse preceding the $t_1$ nutation period leads to a band-selective inversion of the magnetization in the spin-lock frame and will thus invert parts of the nutation spectrum. The position of the inverted part is determined by the modulation frequency $\omega_{mod}$ of

105 the I-BURP-2 pulse and its bandwidth can be adjusted by changing the length of the inversion pulse. Difference spectra can be obtained by combining consecutive scans with and without the inversion pulse and using an appropriate phase cycle on the receiver phase which is shifted by 180°.



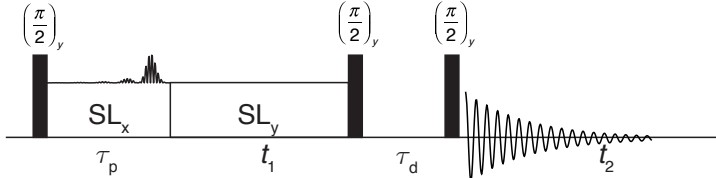

**Figure 3:** Schematic representation of the pulse sequence used for testing of the inversion properties of the I-BURP-2 pulse in the spin-lock frame. After the initial 90° pulse, the magnetization is spin locked along x and the modulated I-BURP-2 inversion pulse applied along y. During the subsequent $t_1$ time the magnetization nutates about the field along y. To obtain pure-phase spectra, a z filter with a dephasing delay is used to select a single component after the nutation. Difference spectra can be obtained by replacing the I-BURP-2 pulse in the spin-lock frame with a simple spinlock in alternating scans while shifting the receiver phase by 180°.

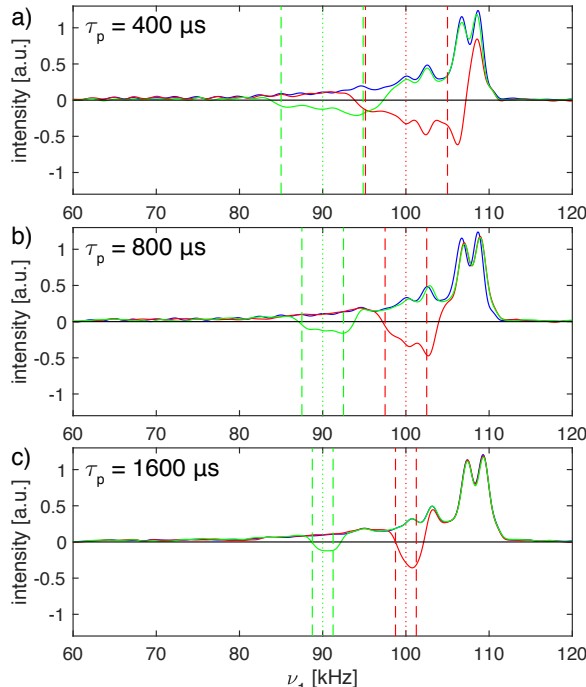

**Figure 4:** Proton nutation spectra of adamantane spinning at 20 kHz at 500 MHz proton resonance frequency using a Bruker 1.9 mm MAS probe. In blue, a standard nutation experiment is shown with a nominal rf-field amplitude of 100 kHz determined by the zero crossing of a 5 μs π pulse. The nutation spectra in green and red were preceded by an I-BURP-2 pulse of length a) 400 μs, b) 800 μs and c) 1600 μs using a modulation frequency of 90 kHz (green) and 100 kHz (red), respectively. The fact that the inversion ranges are slightly shifted compared to the theoretical ones (indicated by dashed and dotted lines) are most likely due to fluctuations in the amplifier gain as can be seen from Fig. S03 in the SI.

Figure 4 shows nutation spectra of adamantane spinning at 20 kHz MAS frequency recorded at a proton resonance frequency of 500 MHz using a Bruker 1.9 mm MAS probe which has a relatively narrow rf-field inhomogeneity profile. The rf-field amplitude corresponds to the zero crossing of a 5 μs π pulse (indicative of a 100 kHz rf-field amplitude) and reaches its maximum around 109 kHz (blue line). Preceding the nutation by an I-BURP-2 pulse in the spin-lock frame inverts part of the nutation spectrum as can be seen from the green and red lines in Fig. 4. The pulses were generated assuming a 100 kHz rf-





field amplitude based on the pulse-length determination, illustrating the good compensation of the I-BURP-2 pulses for

amplitude missetting. The inverted region narrows with increasing pulse length ($\tau_p$ = 400 μs in Fig. 4a, $\tau_p$ = 800 μs in Fig. 4b, $\tau_p$ = 1600 μs in Fig. 4c) and is shifted with a change in the modulation frequency from 90 to 100 kHz (red and green lines in Fig. 4). Virtually no artefacts outside the desired inversion range are visible. At slower spinning frequencies, MAS side bands of the inversion profile become visible (see Fig. S01 of the SI). In comparison to the width and position of the theoretical inversion profiles indicated by dashed (borders) and dotted (centre) lines, the experimental inversion profiles are shifted to

slightly higher rf-amplitudes. A similar experiment on a 600 MHz spectrometer showed no such deviations (Fig. S01 of the SI). This shift is most likely due to fluctuations in the amplifier gain which is shown in detail in Fig. S03 of the SI. Adding a 2 ms spin-lock pulse before the selection pulse (see Fig. S02 for the pulse scheme) leads to a shift of the maximum of the nutation frequency by about 1-2 kHz. This temporal instability of the gain of the radio-frequency amplifiers depends on the exact setting of the radio frequency, the output power of the amplifier and the pulse history. The amplification factor can

slowly drift up or down over time, leading to differences in the selected rf-field amplitudes and the measured ones in the nutation experiment.

We can now combine a nutation spectrum with a preceding band-selective inversion pulse with a nutation spectrum where the I-BURP-2 pulse in the spin-lock frame is replaced with a simple spinlock and the receiver phase is shifted by180° in alternating scans to obtain $B_1$-field selective nutation spectra. Thus, difference spectra are obtained where only certain regions of the rf-

field amplitude are selected by the pulse in the spin-lock frame. This allows the reduction of the rf-field inhomogeneity over the sample at the cost of lowering the signal intensity.  In essence, this is a restriction of the sample not in terms of physical location but in terms of the rf-field experienced by certain crystallites in the rotor.

Figure 5a shows a normal proton nutation spectrum of glycine at a spinning speed of 30 kHz. The spectra were recorded at a proton resonance frequency of 500 MHz using a Bruker 1.9 mm MAS probe. The nominal rf-field amplitude was calibrated

by the zero-crossing of a 5 μs π pulse. In Fig. 5b amplitude-selected difference spectra are overlaid over an expanded region of the nutation spectrum where the modulation frequency of the inversion pulse in the spin-lock frame was shifted through the width of the nutation spectrum.



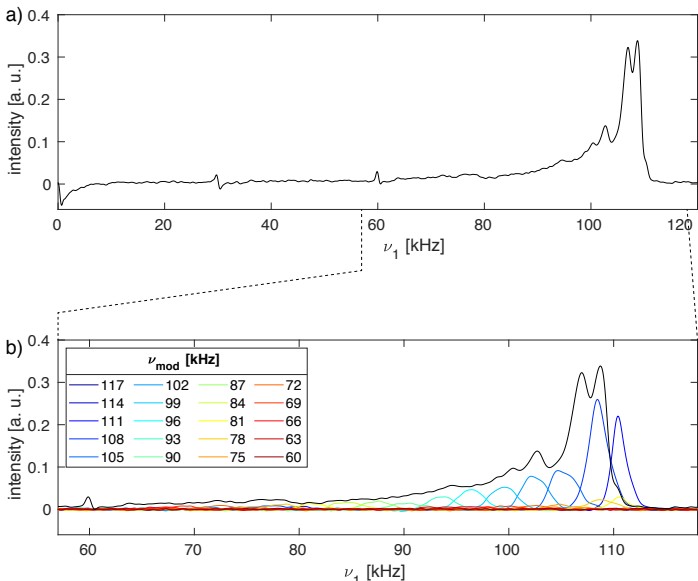

**Figure 5: a)** $^1$H nutation spectra of natural-abundance glycine measured at a proton-resonance frequency of 500 MHz spinning at 30 kHz in
a Bruker 1.9 mm outer-diameter rotor which was completely filled. The nutation spectrum without selective inversion (black) shows MAS
modulation bands at 30 and 60 kHz due to the MAS-induced time dependence of amplitude and phase of the rf irradiation. b) Expanded
region of a) overlaid in colour with various difference spectra using a 2 ms I-BURP-2 pulse in the spin-lock frame where the modulation
frequency was moved through the width of the nutation spectrum. Contributions that lie outside the nutation spectrum are due to the drift of
the radio-frequency amplifier gain over the course of different experiments.

The nutation spectrum without selective inversion (black line) has a maximum at 108 kHz and shows MAS modulation bands

at 30 and 60 kHz that appear due to the MAS-induced time dependence of the rf amplitude and phase. Nutation experiments

with preceding I-BURP-2 pulses with a length of 2 ms and modulation frequencies spanning the width of the nutation spectrum

are shown in colour in Fig. 5b for an expanded region of the nutation spectrum. It can be seen that the experiment allows the

selection of different parts of the rf-field distribution depending on the modulation frequency that is chosen. Interestingly, none

of the sub spectra show the MAS modulation bands at 30 and 60 kHz. As these modulation bands stem from areas within the

sample volume experiencing large rf amplitude and phase modulations, dephasing of the magnetization in those regions during

spinlock could be responsible for their disappearance. However, this is not yet fully understood and requires a more careful

investigation.

To showcase the potential of using rf-field amplitude selective pulses to reduce the effects of rf inhomogeneity in experiments,

homonuclear-decoupled proton spectra under MAS using frequency-switched Lee-Goldburg (FSLG) irradiation (Bielecki et

al., 1989; 1990; Mote et al., 2016) were recorded. FSLG decoupled spectra were acquired in the indirect dimension of a two-

dimensional correlation experiment with direct proton detection under MAS. The pulse sequence used for these experiments

is shown in Fig. 6. All spectra were acquired with the basic FSLG sequence without any super cycling.


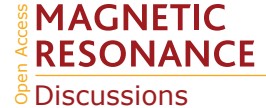

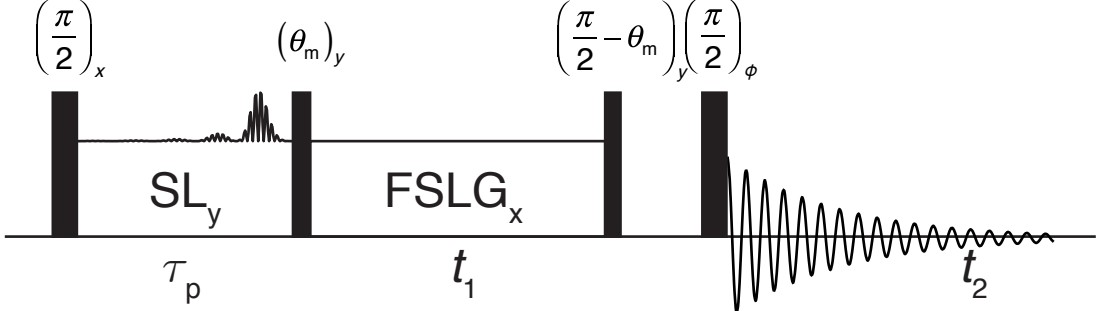


**Figure 6:** Schematic representation of the pulse sequence used to acquire 2D proton-proton chemical-shift correlation spectra with high resolution through FSLG decoupling achieved in $t_1$. The final 90° pulse is phase cycled together with the receiver through all four quadrature phases. States-type phase-sensitive detection in the indirect dimension is implemented by shifting the phase of the first 90° pulse and the spin lock by 90°. To select parts of the sample by the I-BURP-2 pulse in the spin-lock frame, two scans with and without the inversion pulse
were subtracted by shifting the receiver phase by 180°.

Spectra with and without a $B_1$-field selection were acquired at 14 kHz MAS and are shown in Fig. 7 for glycine (Fig. 7a), the

dipeptide β-Asp-Ala (Fig. 7b) and L-histidine (Fig. 7c) at a proton resonance frequency of 500 MHz using a Bruker 1.9 mm

MAS probe. The nominal rf-field amplitude for hard pulses and during spinlock was set to 100 kHz using a nutation experiment

for the calibration. The carrier was placed outside the spectral region of interest, but no experimental optimization of its exact

position was performed. Figure 7 clearly shows that spectra recorded with a rf-field selective 800 μs I-BURP-2 pulse in the

spin-lock frame with a modulation frequency of 100 kHz (red lines) have clearly narrower lines (see Table 1) than FSLG

spectra acquired without the selective inversion (blue lines). Moreover, the zero-frequency artefact at the position of the carrier

frequency is eliminated and the foot on the left side of the peaks due to a distribution of the isotropic chemical-shift scaling

factors (Hellwagner et al., 2020) is reduced. Since the band-selective inversion pulse only selects regions within the sample

space experiencing rf-field amplitudes close to 100 kHz, the selection also leads to a decrease in signal intensity roughly by a

factor of two in the integrated peak intensity of Fig. 7. For some of the peaks, the peak height is reduced significantly less due

to the elimination of the broad components at the left side of the peaks. The experimentally determined chemical shift scaling

factors (see figure caption) are very close to the theoretical value for FSLG decoupling given by $\cos(\theta_m) = \sqrt{1/3} \approx 0.577$.

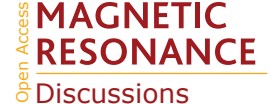

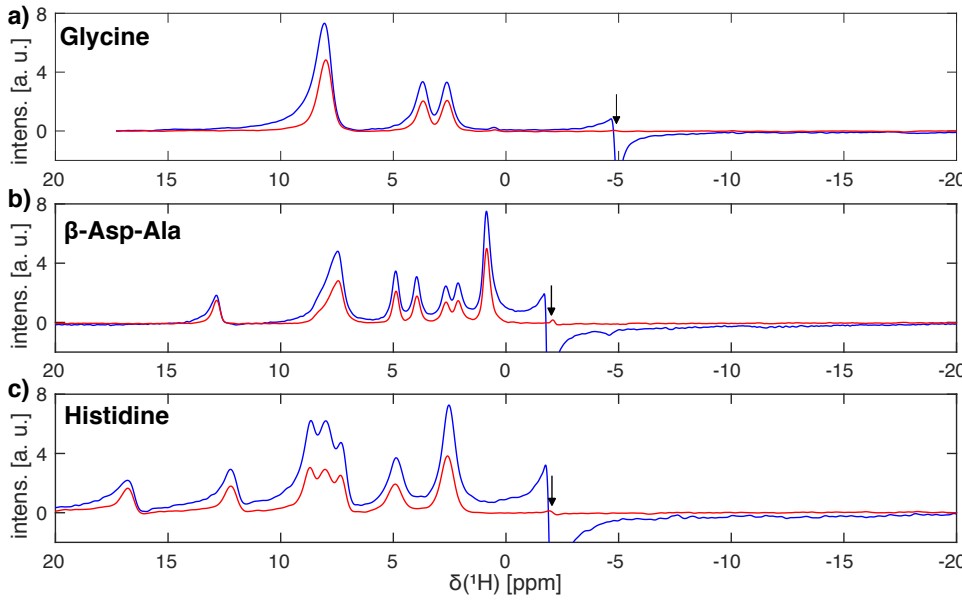


**Figure 7:** Frequency-switched Lee-Goldburg decoupled proton spectra of a) natural-abundance glycine, b) natural-abundance β-Asp-Ala and c) natural-abundance L-histidine with (red) and without (blue) a 800 μs I-BURP-2 pulse used to select the part of the rotor experiencing rf-field amplitudes close to the nominal value of 100 kHz. Spectra were recorded at proton resonance frequency of 500 MHz using a 1.9 mm Bruker MAS probe (completely filled) spinning the sample at 14 kHz. FSLG decoupling was achieved using a continuous phase ramp with

a time resolution of 50 ns and an effective field of 125 kHz. The frequency axes are scaled by the experimentally determined scaling factors of a) 0.562, b) 0.557, c) 0.572) which are very close to the theoretical one of $\cos(\theta_m) = \sqrt{1/3} \approx 0.577$. Arrows indicate the position of the carrier frequency. Narrower lines and a strongly reduced zero-frequency artefact are observed for spectra with a $B_1$-field selection.

The significant reduction of the carrier-frequency artefact permits the recording of spectra where the carrier is positioned in the centre of the spectral region of interest. This allows a substantial reduction of the spectral width and faster data acquisition

in the 2D schemes used here. Resulting FSLG decoupled spectra of L-histidine with different pulse lengths for the I-BURP-2 selection pulse acquired at a proton resonance frequency of 600 MHz using a 2.5 mm Bruker MAS probe spinning the sample at 14 kHz are shown in Fig. 8. Increasing the length of the I-BURP-2 pulse and thus improving the rf-field selectivity (narrower bandwidth) leads to a decrease in linewidth and a reduction of the foot of the lines pointing away from the carrier frequency which is due to the distribution of isotropic chemical-shift scaling factors (Hellwagner et al., 2020). However, the achieved

line narrowing using more selective pulses again coincides with lower signal intensity (see Table 2 for numerical values of the line width). The intensity of the signals drops significantly by a factor of two to three when going from 200 or 400 μs pulse length to a much more selective 800 μs or 2 ms pulse length. Figure 8 also illustrates that the use of $B_1$-field selective pulses allows the placement of the carrier frequency inside the region of interest, as only a small carrier frequency artefact is observed which can be completely eliminated by placing the carrier outside the region of interest (Fig. 8, blue spectrum). Again, the

chemical-shift scaling factors (see figure caption for numerical values) are very close to the theoretical value.



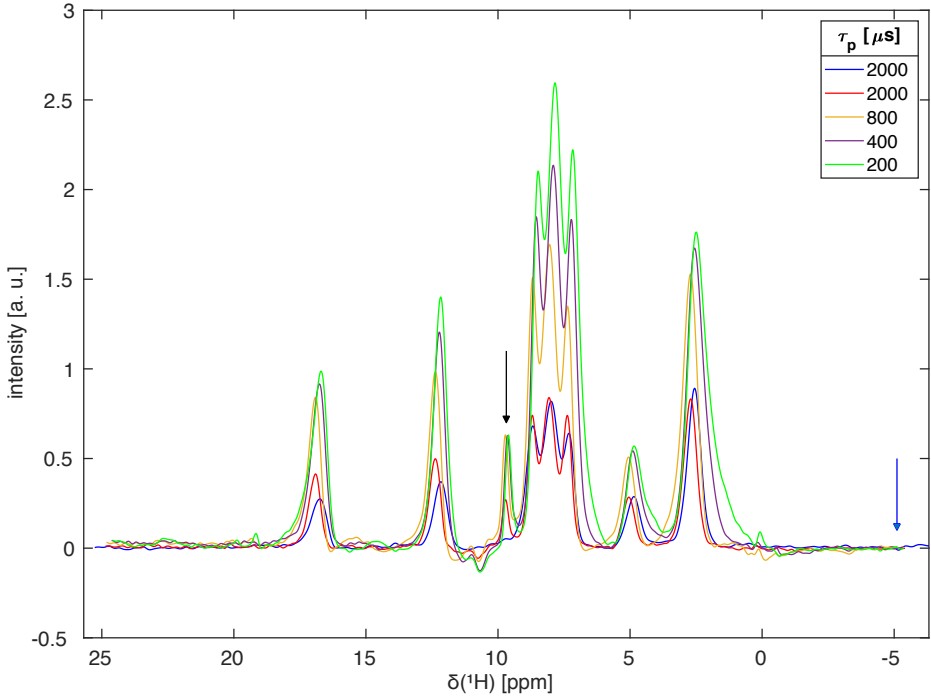

**Figure 8:** Homonuclear-decoupled proton spectra of L-histidine using FSLG decoupling and different selectivity of the I-BURP-2 pulse in the spin-lock frame at 14 kHz MAS. The spectra were acquired at a proton resonance frequency of 600 MHz using a Bruker 2.5 mm probe which has a much stronger rf-field inhomogeneity than the 1.9 mm MAS probe used in Fig. 7. The carrier was placed in the centre of the region of interest (arrow indicating the exact position) for all spectra except the blue one. One can clearly see that longer (more selective) pulses reduce the intensity of the signal but lead to narrower lines especially on the side of the peak pointing away from the carrier frequency. There is a small carrier-frequency artefact for the spectra with the carrier frequency inside the spectral region of interest. Experimental scaling factors were 0.572 for the 2 ms I-BURP-2 pulses, 0.575 for the 800 µs pulse and 0.579 for the 400 and 200 µs pulses. The modulation frequency was set to 100 kHz for the 2 ms and 800 µs, 95 kHz for the 400 µs, and 90 kHz for the 200 µs I-BURP-2 pulses.






**Table 1:** Full width at half maximum (FWHM) for homonuclear decoupled proton spectra at 500 MHz shown in Figure 7.

| Sample | Figure | δ(¹H) [ppm] | FWHM[1] (standard FSLG) | | FWHM[1] (800 µs I-BURP-2) | |
|---|---|---|---|---|---|---|
| Glycine | 7a) | 2.62 | 165 Hz | 0.64 ppm | 162 Hz | 0.58 ppm |
| | | 3.68 | 192 Hz | 0.68 ppm | 165 Hz | 0.59 ppm |
| | | 8.00 | 241 Hz | 0.86 ppm | 204 Hz | 0.73 ppm |
| β-Asp-Ala | 7b) | 0.86 | 113 Hz | 0.41 ppm | 92 Hz | 0.33 ppm |
| | | 3.95 | 113 Hz | 0.41 ppm | 95 Hz | 0.34 ppm |
| | | 4.88 | 104 Hz | 0.37 ppm | 89 Hz | 0.32 Hz |
| | | 7.45 | 259 Hz | 0.93 ppm | 217 Hz | 0.78 ppm |
| | | 12.84 | 134 Hz | 0.48 ppm | 104 Hz | 0.37 ppm |
| L-Histidine | 7c) | 2.55 | 214 Hz | 0.74 ppm | 198 Hz | 0.69 ppm |
| | | 4.91 | 256 Hz | 0.89 ppm | 217 Hz | 0.76 ppm |
| | | 12.21 | 263 Hz | 0.92 ppm | 191 Hz | 0.67 ppm |
| | | 16.79 | 374 Hz | 1.31 ppm | 198 Hz | 0.69 ppm |

[1] Values given in Hz are taken from the processed and unscaled spectra. Values in ppm include the isotropic chemical-shift scaling and correspond to the values in the plotted spectra.


**Table 2:** Full width at half maximum (FWHM) for homonuclear decoupled proton spectra of L-Histidine at 600 MHz shown in Figure 8.

| δ(¹H) [ppm] | FWHM[1] $\tau_p = 200$ µs | | FWHM[1] $\tau_p = 400$ µs | | FWHM[1] $\tau_p = 800$ µs | | FWHM[1] $\tau_p = 2000$ µs (centre) | | FWHM[1] $\tau_p = 2000$ µs (edge) | |
|---|---|---|---|---|---|---|---|---|---|---|
| 2.55 | 300Hz | 0.86 ppm | 242 Hz | 0.75 ppm | 226 Hz | 0.66 ppm | 201 Hz | 0.58 ppm | 203 Hz | 0.59 ppm |
| 4.91 | 249Hz | 0.72 ppm | 224 Hz | 0.64 ppm | 198 Hz | 0.58 ppm | 183 Hz | 0.53 ppm | 209 Hz | 0.61 ppm |
| 12.21 | 191Hz | 0.55 ppm | 181 Hz | 0.52 ppm | 178 Hz | 0.52 ppm | 173 Hz | 0.50 ppm | 203 Hz | 0.59 ppm |
| 16.79 | 242 Hz | 0.70 ppm | 219 Hz | 0.63 ppm | 193 Hz | 0.56 ppm | 181 Hz | 0.52 ppm | 234 Hz | 0.68 ppm |

[1] Values given in Hz are taken from the processed and unscaled spectra. Values in ppm include the isotropic chemical-shift scaling and correspond to the values in the plotted spectra.




## 5. Materials and Methods

Numerical simulations were implemented using the spin-simulation environment GAMMA (Smith et al., 1994). All simulations were performed using a single-spin system and time-slicing of the time-dependent Hamiltonian to calculate the time evolution of the density operator in the standard rotating frame.

All powdered samples used in the measurements (adamantane, natural-abundance glycine, natural-abundance L-histidine·HCl·H$_2$O, natural-abundance β-Asp-Ala) were purchased from commercial sources and used without further purification. Rotors were filled completely and the sample space was not spatially restricted.

Experiments were carried out on Bruker Avance III HD NMR spectrometers, operating at a proton resonance frequency of 500 MHz (600 MHz), equipped with a Bruker 1.9 mm (2.5 mm) triple-resonance probe (in double resonance configuration) at

a temperature of 285 K. The shaped pulses for the FSLG homonuclear decoupling and the I-BURP-2 pulses were programmed in Matlab (The Mathworks, Natick, MA) and exported to Bruker TopSpin shape file format using home written routines.

Two-dimensional nutation spectra were acquired with a simple cosine modulation in $t_1$ at a spinning speed of 30 kHz (20 kHz) for adamantane (glycine) at a proton resonance frequency of 500 MHz. The spinlock rf-field amplitude used for the I-BURP-2 pulses was set to 100 kHz as determined by the zero-crossing of a 5 μs π pulse. Typically, 256-512 $t_1$ increments with eight

scans each and a time increment of 3.5 μs, corresponding to a spectral width of 286 kHz, were recorded. The spectral width in $t_2$ was set to 100 kHz and 1024 complex data points were acquired. Spectra were processed in Matlab using a cosine-squared apodization function. Example processing scripts are given in the SI.

Homonuclear decoupled proton spectra were acquired as 2D $^1$H-$^1$H correlation spectra with FSLG decoupling (Bielecki et al., 1989; 1990; Mote et al., 2016) in the indirect dimension using States-type data acquisition (States et al., 1982) for sign

discrimination and phase-sensitive detection in the indirect dimension. Measurements were performed at proton resonance frequencies of 500 and 600 MHz at a spinning speed of 14 kHz. The rf-field amplitudes used for hard pulses and during spinlock were set to 100 kHz as determined using a nutation experiment. Shaped pulses using a phase ramp with 50 ns time resolution were used for the implementation of FSLG decoupling. Each shape consists of 320 points leading to a total length of 16 μs which corresponds to a nutation frequency about the effective field of 125 kHz. Spectra were recorded with 350 - 768

$t_1$ increments with eight to 16 scans each and a time increment between 48 and 96 μs (spectral width 20.8 to 10.4 kHz). The spectral width in $t_2$ was set to 200 kHz and 1024 complex data points were acquired. For most experiments, the carrier was set to the edge of the spectral region of interest, but no experimental optimization of the carrier position was performed. In all spectra, the carrier position is indicated by an arrow. A more detailed summary of the acquisition parameters used for the homonuclear decoupled proton spectra can be found in Table 1 in the SI. For processing in Matlab, zero filling to 4096x4096

data points was used and a cosine-squared window function applied. The 1D spectra were obtained by summing over the relevant spectral region in $\omega_2$. Experimental chemical shift scaling factors were determined by comparing the observed peak positions to those found in the literature. The reference peaks used for the comparison were the NH$_3$ resonance and the centre of the methylene resonance (8 ppm and 3.16 ppm) for glycine (Mote et al., 2016), the OH and CH$_3$ resonances (12.85 ppm and

**MAGNETIC RESONANCE** Discussions Open Access

0.86 ppm) for β-Asp-Ala (Paruzzo et al., 2019) and the α and $\delta^2$ proton resonances (16.75 ppm and 2.55 ppm) for L-histidine·HCl·H₂O (Mithu et al., 2013).

## 6 Conclusions and Outlook

We have shown that band-selective pulses in a spin-lock frame can be used to implement nutation-frequency selective pulses with very good excitation profiles. Implementing nutation-frequency selective pulses in this way offers a large range of possible pulse shapes and leads to more stable results than direct numerical optimization of these pulses described in the literature thus far (Charmont et al., 2002). Such $B_1$-field selective pulses can be used to reduce detrimental effects in pulse sequences that are sensitive to rf-field inhomogeneity and present a much simpler and more effective alternative to spatial sample restriction along the rotor axis. As an example, we show significant improvements in homonuclear-decoupled spectra under MAS using FSLG decoupling. On one hand, the zero-frequency artefact is almost eliminated and the lines are narrower due to the reduction of the rf-field inhomogeneity. At the same time, the intensity of the signal intensity is reduced and a compromise between resolution and sensitivity has to be found. We believe that pulse lengths around 500 μs and 1 ms (bandwidth of 4 to 8 kHz) provide a good compromise. The implementation of the pulses is straightforward and any inversion pulse shape can be used. In principle selective 90° pulses or more complex composite pulses could also be used to implement a selection of $B_1$-fields or other manipulations of the magnetization in the spin-lock frame. Moreover, nutation-frequency selective pulses could also be used for probe background suppression (Feng and Reimer, 2011) in a simple difference experiment since the probe background typically experiences much lower rf-field amplitudes than the sample inside the coil.

**Data availability.**

Pulse-shape files, experimental data, and simulations are available upon request from the corresponding author.

**Supplement.**

The supplement related to this article is available online at:

**Author contributions.**

NW and ME designed the research. ZT provided data about rf-field inhomogeneity in probes. KA and ME carried out all measurements and simulations. All authors were involved in writing the manuscript.



**Competing interests.**

The authors declare that they have no conflict of interest.

**Acknowledgements.**

We would like to thank Perunthiruthy K. Madhu, Kaustubh Mote, and Johannes Hellwagner for insightful discussions about theory and experimental implementation of homonuclear decoupling. Beat H. Meier and Alexander Barnes are acknowledged for providing measurement time for the project.

**Financial support.**

This research has been supported by the Schweizerischer Nationalfonds zur Förderung der Wissenschaftlichen Forschung (grant no. 200020_188988 (ME)) and the Czech Science Foundation (grant no. 20-00166J (ZT)).

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
