# Peer review of "Using nutation-frequency-selective pulses to reduce radio-frequency field inhomogeneity in solid-state NMR"

_Magnetic Resonance, 2020_

## Referee Comment (RC1) · Anonymous Referee #1 · 30 Jul 2020

The manuscript presents an rf-amplitude-selective inversion method in the spin-lock rotating frame. This method enables spin inversion of an selective area in the sample experiencing a narrow-band of B1 (inhomogeneous) rf field. After the Introduction, the Theory of the B1 selectivity is presented and demonstrated for I-BURP 2 inversion in Fig. 1. To assist the reader, it would have been helpful to add to Fig 1a "the doubly rotating frame", to 1c and 1d-e "the spin lock rotating frame". I understand that in Eq. 2 phi(t) does not contain the counter-rotator terms and therefore what is the meaning of "Therefore" on line 51. Eq. 3 assumes that w0-wfr=0. Numerical results show the inversion selectivity and indicates the correlation between the length of the inversion pulse and the band selectivity. Experiment-1 demonstrates the inversion  to <-Ix>

beautifully via a nutation experiment around a y-spin lock field. Experiment-2 repeats this experiment on glycine instead of on adamanate. The results are compare with a standard spin-lock preparation. Unfortunately, the color coding of the results shown in Fig 5 are not sufficient to appreciate the success of the experiments. The "blue" curve doesn't have an oscillating frequency dependence, but is rather composed of many profiles. Furthermore, the "red" lines are indistinguishable. Experiment-3 presents an implementation of the selective inversion method. FSLG decoupled spectra of L-histidine show very nicely all expected effects; line narrowing and line intensity loss. Overall I thus recommend publication with some minor modification.

---

## Referee Comment (RC2) · Paul Hodgkinson (Referee) · 5 Aug 2020

This is an excellent contribution to the difficult problem of understanding and optimising $^1$H homonuclear decoupling in solid-state NMR.

The introduction gives an effective overview of prior work. I would add the method proposed by Odedra and Wimperis (DOI: 10.1016/j.jmr.2013.04.002) for quick-and-dirty imaging of the RF imhomogeneity using the $z$ shim (i.e. not needing a magic-angle gradient). I feel it should be clearer that this work builds, conceptually at least, on the 2002 Charmont paper cited. This can be done by moving the sentence beginning "Alternatively" to the start of the final paragraph.

The theory is mostly clearly explained, although some wording is a bit unclear which meant I was reading some sentences a couple of times to understand the point being made. For example, we don't really "apply RF orthogonal to the static magnetic field" (at least not in MAS probes). The truncation of the Hamiltonian at high field could be more carefully expressed, particularly as this is important for the following step. I similarly didn't understand the term "Larmor-frequency axis" (a frequency can't have an axis). I wasn't entirely convinced by the analogy between the spin-lock frame and the (Larmor) rotating frame; it is clear that $\omega_1$ is much smaller than the Larmor frequency, but much less obvious, e.g. Fig. 1(c), that $\omega_2$ is much less than $\omega_1$. The breakdown of this approximation is referred to, but it is not clear how this would manifest itself in practice. Wouldn't it show in Fig. 2 as deviations between the simulated profile with and without the spin-lock component?

In terms of the simulations, I didn't really understand the significance of the blue line. Is this trying to approximate the effects of RF imhomogeneity without assuming a particular RF profile?

The experiments are all clearly described. The disappearance of the modulation sidebands when using the selective excitation pulse (Fig. 5) is interesting. Could this simply be explained by the breaking the rotor cycle periodicity by the time-dependent (and unsynchronised) shaped pulse?

The results obtained from the two different probes (Figs 7 and 8) were strikingly different and I feel deserve more comment. In particular, the response of the zero-frequency artifacts is very different. The artifact is much larger in Fig. 7, but vanishes when the shaped pulse is used, whereas the artifact is much smaller in Fig. 8 and the changes are much more modest. Is the behaviour in Fig. 7 reproducible? It is not obvious why the selective excitation would strongly reduce zero-frequency artifacts, and given that this is highlighted in the conclusions, it would good to understand this point a little better. There is some repetition in this section, e.g. Hellwagner 2020 doesn't need to be cited twice to make the same point. (Similarly for Bloch-Seigert earlier.)

Figure 8 is quite hard to read. It is not clear why different modulation frequencies were used for different shaped pulse lengths.

I'm not convinced that the linewidth data in Tables 1 and 2 is that useful in the main text, since I doubt that the quantitative results will be very reproducible - certainly not to 1 Hz! For example, there is a large reduction in FWHM specifically for the 16.8 ppm peak in Fig. 7, while the peaks behave much more uniformly in Fig. 8. The width of the 16.8 ppm in Fig. 7 looks more like a phase artifact (perhaps associated with the homonuclear decoupling?) rather than a tail due to RF imhomogeneity. The interacting nature of effects is a major headache for these experiments, but this is why a more detailed discussion of the weight that can be put on the quantitative results would be helpful.

It's good that the data sets are available on request. It would be even better if they were available via an open data repository!

Minor typographical issues:

- Not all symbols are defined, e.g. is $\tau_p$ in Fig 3 the duration of the shaped pulse or the combined duration of spin-lock + shaped pulse prior to nutation measurement?

- Axes are conventionally italic, e.g. $x$.

- In line 18, there should probably be a comma after "NMR". As currently worded, the sentence implies that there are some MAS probes in which the coil is not close to the sample. Adding the comma creates a separate clarifying clause which applies to all small MAS probes (probably the intended meaning).

Paul Hodgkinson

---

## Author Comment (AC1) · 13 Aug 2020

*To assist the reader, it would have been helpful to add to Fig 1a "the doubly rotating frame", to 1c and 1d-e "the spin lock rotating frame".

All plots shown in Figure 1 are in the "usual rotating frame", i.e. in the frame where we rotate with the Larmor frequency around the z axis. Figure 1a shows the amplitude and 1b the modulation frequency while 1c-1e show the combined modulation in different representations (x/y) and (amplitude/phase). But they are all in the rotating frame. Therefore, we did not add the suggested labels.

*I understand that in Eq. 2 phi(t) does not contain the counter-rotator terms and therefore what is the meaning of "Therefore" on line 51. Eq. 3 assumes that w0-wfr=0.

We have rephrased this part to make this clearer and changed "Therefore, we obtain a first-order ..." to "Neglecting the counter-rotating part, we obtain a first-order ..." and also added the on-resonance condition "Assuming $\omega\_0=\omega\_rf$" before Eq. 3.

*Unfortunately, the color coding of the results shown in Fig 5 are not sufficient to appreciate the success of the experiments. The "blue" curve doesn't have an oscillating frequency dependence, but is rather composed of many profiles. Furthermore, the "red" lines are indistinguishable

We do not see a simple solution to this problem since we want to show all the sub spectra in a single plot. We have decided to add a Figure to the SI which shows all the spectra in a stacked way so that they can be viewed independently. We added a sentence to the caption: "To allow a more detailed view of the various sub spectra in b), we show them in a stacked way in Fig. S04 of the SI."

We would like to thank the reviewer for carefully reading the manuscript and for his suggestions and hope that these changes address the concerns of reviewer 1 adequately.

―――――――――――――――――――

[Figure]

**Fig. 1.** New Figure S04

---

## Author Comment (AC2) · 13 Aug 2020

*The introduction gives an effective overview of prior work. I would add the method proposed by Odedra and Wimperis (DOI: 10.1016/j.jmr.2013.04.002) for quick-and-dirty imaging of the RF inhomogeneity using the z shim (i.e. not needing a magic-angle gradient). I feel it should be clearer that this work builds, conceptually at least, on the 2002 Charmont paper cited. This can be done by moving the sentence beginning "Alternatively" to the start of the final paragraph.

We have added the reference to the work by Odedra and Wimperis on line 21: " ... but also simpler methods using the z shim have been proposed {Odedra:2013dw}". In

addition, we have changed the breaking of the paragraphs to emphasize the Charmont paper more clearly.

*The theory is mostly clearly explained, although some wording is a bit unclear which meant I was reading some sentences a couple of times to understand the point being made. For example, we don't really "apply RF orthogonal to the static magnetic field" (at least not in MAS probes). The truncation of the Hamiltonian at high field could be more carefully expressed, particularly as this is important for the following step. I similarly didn't understand the term "Larmor-frequency axis" (a frequency can't have an axis).

There are many textbooks that discuss in detail the Bloch-Siegert shift, wo we decided to only show the static part and neglect the counter rotating part in Eq. (2). However, we have made this clearer by changing the sentence before Eq. (2) to: " Neglecting the counter-rotating part, we obtain a ...". We have rephrased the (admittedly awkward expression) "Larmor-frequency axis" to: "Using an appropriate phase or amplitude modulation will allow us to implement frequency band-selective pulses in the normal rotating frame {Emsley:2007ej}."

*I wasn't entirely convinced by the analogy between the spin-lock frame and the (Larmor) rotating frame; it is clear that $\omega 1$ is much smaller than the Larmor frequency, but much less obvious, e.g. Fig. 1(c), that $\omega 2$ is much less than $\omega 1$. The breakdown of this approximation is referred to, but it is not clear how this would manifest itself in practice. Wouldn't it show in Fig. 2 as deviations between the simulated profile with and without the spin-lock component?

Originally, we expected that the Bloch-Siegert effect would be very important in this situation where the amplitude of the IBURP pulse is less than an order of magnitude lower than the spin-lock amplitude. However, numerical simulations (as in Fig. 2) using a circular-polarized field show almost identical profiles as in Fig. 2 and zero inversion if the counter-rotating circular-polarized field is used. We have added a figure to the

SI (Fig. S05) that shows these simulations and refer to these simulations in the figure caption of Fig. 2. We believe that the main reason for not seeing any significant Bloch-Siegert effects is the fact that the second-order Bloch-Siegert terms will point in the direction of the spin-lock axis and lead only to a change in the magnitude of the spin lock. Since the pulses have an almost square excitation profile, such additional Bloch-Siegert fields will only have a significant effect at the edges. We have added a sentence to the figure caption of Fig. 2: " Simulations using circular-polarized radio-frequency fields that address the role of possible Bloch-Siegert effects can be found in Fig. S05 of the SI."... while the blue line uses $\nu 2$ = ... a s it would be the case in a real experiment with rf-field inhomogeneity."

*In terms of the simulations, I didn't really understand the significance of the blue line. Is this trying to approximate the effects of RF inhomogeneity without assuming a particular RF profile?

For the black line, the IBURP pulse has always the ideal rf-field amplitude and only the spin-lock changes. For the blue line, the amplitude of the IBURP pulse is assumed to be proportional to the spin-lock amplitude as it would happen in a real experiment with inhomogeneity. This is explained in the figure caption. We have extended the last sentence in the figure caption of Fig. 2 to read: "

*The experiments are all clearly described. The disappearance of the modulation side-bands when using the selective excitation pulse (Fig. 5) is interesting. Could this simply be explained by the breaking the rotor cycle periodicity by the time-dependent (and unsynchronised) shaped pulse?

The modulation side bands also disappear under a spin lock without a pulse or they would be present in the difference spectra. We do not fully understand this yet as it is also stated in the main text and would prefer not to speculate more about this.

*The results obtained from the two different probes (Figs 7 and 8) were strikingly different and I feel deserve more comment. In particular, the response of the zero-frequency

artifacts is very different. The artifact is much larger in Fig. 7, but vanishes when the shaped pulse is used, whereas the artifact is much smaller in Fig. 8 and the changes are much more modest. Is the behaviour in Fig. 7 reproducible? It is not obvious why the selective excitation would strongly reduce zero-frequency artifacts, and given that this is highlighted in the conclusions, it would good to understand this point a little better. There is some repetition in this section, e.g. Hellwagner 2020 doesn't need to be cited twice to make the same point. (Similarly for Bloch-Siegert earlier.)

This must be a misunderstanding. In Fig. 7, we compare PMLG spectra with and without rf-amplitude selection and the zero-frequency artefact is strong in PMLG without selection of the amplitude. In Fig. 8 all spectra are with rf-amplitude selection and show very small artefacts. The artefacts are slightly bigger if we put the carrier in the center. We believe that the zero-frequency artefact comes from parts of the rotor where the rf field is much lower than the desired amplitude and these parts are suppressed by the selection. We hope this clarifies this question.

*Figure 8 is quite hard to read. It is not clear why different modulation frequencies were used for different shaped pulse lengths.

The different modulation frequencies were chosen to make sure that the pulses of different band width are centered in the region of maximum intensity of the nutation spectrum. This requires a lower modulation frequency for pulses with larger bandwidth. We have added the following sentence to Fig. 8: " The modulation frequencies were selected such that the band widths of the different pulses cover the region of maximum intensity in the nutation spectrum."

I'm not convinced that the linewidth data in Tables 1 and 2 is that useful in the main text, since I doubt that the quantitative results will be very reproducible - certainly not to 1 Hz! For example, there is a large reduction in FWHM specifically for the 16.8 ppm peak in Fig. 7, while the peaks behave much more uniformly in Fig. 8. The width of the 16.8 ppm in Fig. 7 looks more like a phase artifact (perhaps associated with the

homonuclear decoupling?) rather than a tail due to RF inhomogeneity. The interacting nature of effects is a major headache for these experiments, but this is why a more detailed discussion of the weight that can be put on the quantitative results would be helpful.

We believe that the line width data illustrate the line narrowing by restricting the rf-field inhomogeneity quite nicely but we agree that a three-digit precision is not warranted. We have rounded the line width data to 10 Hz and we have also corrected the phasing problems in 7c that lead to the too large line width. Thanks for catching this. We have also rechecked the rest of the line width data and updated the table. Figure 8 shows only data with rf-field selection and illustrates the narrowing associated with longer selection pulses (narrower rf-field distribution).

*It's good that the data sets are available on request. It would be even better if they were available via an open data repository!

We will consider making the data publicly available in a repository.

Minor typographical issues:

*Not all symbols are defined, e.g. is $\tau$p in Fig 3 the duration of the shaped pulse or the combined duration of spin-lock + shaped pulse prior to nutation measurement?

The duration $\tau$p is the length of the IBURP pulse and at the same time the length of the spin lock since the spin lock was always adjusted to the length of the pulse.

*Axes are conventionally italic, e.g. x.

We have tried to change all axes labels in figures and in the text to italic. I hope we did not miss any of them.

*In line 18, there should probably be a comma after "NMR". As currently worded, the sentence implies that there are some MAS probes in which the coil is not close to the sample. Adding the comma creates a separate clarifying clause which applies to all

small MAS probes (probably the intended meaning).

This is indeed correct. We added the comma in the revised text.

We would like to thank Paul Hodgkinson for carefully reading the manuscript and for his suggestions which, we believe, improved the paper. We hope that these changes address all the issues raised.

[Figure]

[Figure]

**Fig. 1.** New SI Figure S05 top row is linear vs. circular (+) and bottom row is linear vs. circular (-) polarized rf.

---

## Referee Comment (RC3) · Paul Hodgkinson (Referee) · 17 Aug 2020

That looks fine to me. A couple of quick notes:

One of the replies has been truncated: "For the black line, the IBURP pulse has always the ideal rf-field amplitude [...]. We have extended the last sentence in the figure caption of Fig. 2 to read: " [Added text is missing, but it will be in the manuscript.]

"This must be a misunderstanding. In Fig. 7, we compare PMLG spectra with and without rf-amplitude selection and the zero-frequency artefact is strong in PMLG without selection of the amplitude. In Fig. 8 all spectra are with rf-amplitude selection and

show very small artefacts. The artefacts are slightly bigger if we put the carrier in the center. We believe that the zero-frequency artefact comes from parts of the rotor where the rf field is much lower than the desired amplitude and these parts are suppressed by the selection. We hope this clarifies this question."

This wasn't really a misunderstanding, more a reservation over the artifacts being described as "very small" in Fig. 8. The suppression is excellent off resonance in Fig. 7 and 8 (essentially complete suppressed), but only moderate on resonance. I guess the query is more over on vs. off-resonance behaviour.

PH

---

## Author Comment (AC3) · 18 Aug 2020

*That looks fine to me. A couple of quick notes: One of the replies has been truncated: "For the black line, the IBURP pulse has always the ideal rf-field amplitude [...]. We have extended the last sentence in the figure caption of Fig. 2 to read: " [Added text is missing, but it will be in the manuscript.]

The sentence was actually in the reply but accidentally pasted one section earlier. We have extended the last sentence in the figure caption of Fig. 2 to read: "... while the blue line uses $\nu 2 = ...$ as it would be the case in a real experiment with rf-field inhomogeneity." This will be corrected in the final response.

[Figure]

*"This must be a misunderstanding. In Fig. 7, we compare PMLG spectra with and without rf-amplitude selection and the zero-frequency artefact is strong in PMLG without selection of the amplitude. In Fig. 8 all spectra are with rf-amplitude selection and show very small artefacts. The artefacts are slightly bigger if we put the carrier in the center. We believe that the zero-frequency artefact comes from parts of the rotor where the rf field is much lower than the desired amplitude and these parts are suppressed by the selection. We hope this clarifies this question." This wasn't really a misunderstanding, more a reservation over the artifacts being described as "very small" in Fig. 8. The suppression is excellent off resonance in Fig. 7 and 8 (essentially complete suppressed), but only moderate on resonance. I guess the query is more over on vs. off-resonance behaviour.

Sorry for our misunderstanding of the original comment. This is a valid point. The axial peak with rf-field selection gets bigger, the closer the carrier gets to an area with spectral intensity. We are not really sure why this happens. However, compared to the axial peak without a B1-field selective pulse (see Fig. 7), the remaining axial peaks in Fig. 8 are still quite small but the suppression is not as perfect as it is outside the spectral range. We have amended the text to read now: "Figure 8 also illustrates that the use of B1-field selective pulses allows the placement of the carrier frequency inside the region of interest, as only a small but clearly visible carrier-frequency artefact is observed which can be completely eliminated by placing the carrier outside the region of interest (Fig. 8, blue spectrum). The magnitude of the carrier-frequency peak increases with increasing proximity to a real spectral peak." Unfortunately, we did not measure a spectrum without the selection pulse and the carrier in the center of the spectrum since they are usually dominated by the zero-frequency peak.